

# Probing the particle spectrum of nature with evaporating black holes

Michael J. Baker[1*] and Andrea Thamm[2†]

**1** ARC Centre of Excellence for Dark Matter Particle Physics, School of Physics,
The University of Melbourne, Victoria 3010, Australia
**2** School of Physics, The University of Melbourne, Victoria 3010, Australia

⋆ michael.baker@unimelb.edu.au , † andrea.thamm@unimelb.edu.au

## Abstract

Photons radiated from an evaporating black hole in principle provide complete information on the particle spectrum of nature up to the Planck scale. If an evaporating black hole were to be observed, it would open a unique window onto models beyond the Standard Model of particle physics. To demonstrate this, we compute the limits that could be placed on the size of a dark sector. We find that observation of an evaporating black hole at a distance of 0.01 parsecs could probe dark sector models containing one or more copies of the Standard Model particles, with any mass scale up to 100 TeV.

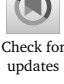
# 1 Introduction

Determining the particle spectrum of nature is one of the fundamental goals of physics. The last 120 years have seen a huge advance in our understanding of the elementary particles, from J.J. Thomson's discovery of the electron in 1897 [1] to the discovery of the Higgs boson at CERN in 2012 [2,3], completing the Standard Model (SM) of particle physics.

The last 100 years have also seen a huge advance in our understanding of black holes (BH), from Schwarzschild [4] and Droste's [5] exact solutions to the Einstein field equations, which would prove to describe the simplest black holes, in 1916 to the 2016 observation of gravitational waves from a binary black hole merger by the LIGO and Virgo Collaborations [6]. This was quickly followed by the first direct image of a black hole by the Event Horizon Telescope [7].

In 1974 Hawking [8,9] combined arguments from quantum mechanics and general relativity to predict that black holes should radiate particles, so-called Hawking radiation, and lose mass. The emission is approximately black-body, with a temperature that is inversely proportional to the black hole's mass. As the black hole radiates, it loses mass and heats up, leading to a runaway evaporation process. While the solar mass and supermassive black holes already observed will not evaporate any time soon, primordial black holes with masses around $10^{15}$ g, which may have been produced in the early universe [10–33], would be evaporating today (see, e.g., refs. [34–39] for recent reviews of primordial black holes). Although there is not yet any clear evidence of evaporating black holes (EBHs), they have been invoked to explain, e.g., fast gamma ray bursts [40], antimatter in cosmic rays [41–43], and the galactic gamma ray background [44].

Evaporating black holes predominantly radiate all elementary particles with a mass less than their temperature. When the temperature rises above a particle mass threshold, a new radiation process becomes unsuppressed, the black hole loses mass at a faster rate, and the temperature increases at a faster rate. This continues until the temperature reaches the Planck scale, at which point quantum gravity effects may become important. Since photons are massless they are always emitted by evaporating black holes, with an energy similar to the black hole temperature. In addition, other radiated particles may also produce photons after their emission. In this way, the photon signal from an evaporating black hole encodes detailed information about the evaporation rate and the complete particle spectrum of nature.

Experiments such as the HAWC Observatory are actively searching for evaporating black holes. In this work we consider what information could be obtained from an observation in practice, and the extent to which Beyond the Standard Model (BSM) scenarios could be probed. As an illustrative scenario we consider dark sector models. Dark sector models are strongly motivated by the observation of dark matter, but at present there are no known general probes of the extent of the dark sector.

While the impact of non-Standard Model physics on black hole evaporation has been discussed in the literature, this has predominantly focused on Hagedorn-type models [45], e.g. refs. [40,46], which have now been superseded by quantum chromodynamics or BSM particle production, e.g. refs. [47–66]. In contrast, the extent to which the parameter space of contemporary BSM models could be probed by the observation of an evaporating black hole is relatively unexplored (we are only aware of ref. [67], which contains a limited analysis in the case of a single squark).

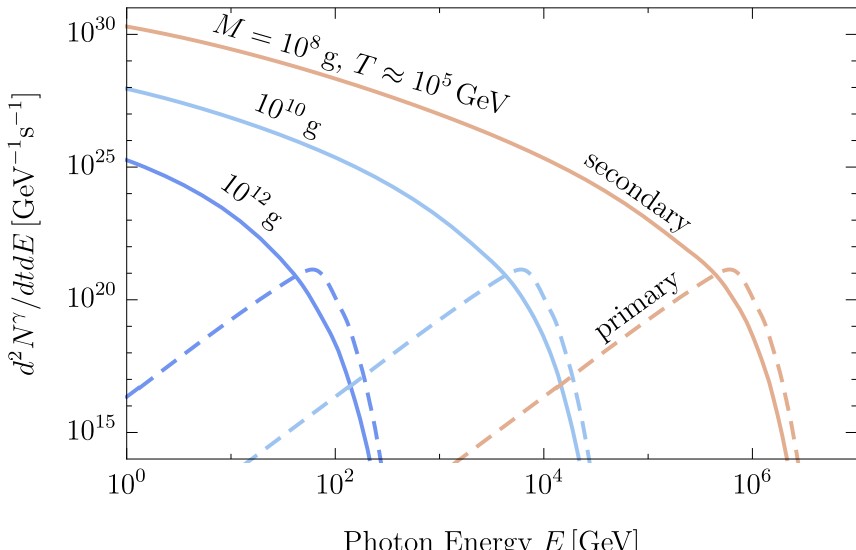

Figure 1: The primary (dashed) and secondary (solid) photon spectra at BH masses $M = (10^{12}, 10^{10}, 10^8)\,$g. In the SM, this corresponds to $\tau = (5 \times 10^8, 4 \times 10^2, 4 \times 10^{-4})\,$s where $\tau$ is the remaining lifetime of the EBH.

## 2 Formalism

We now discuss the theoretical framework of BH evaporation, calculate the resulting photon spectra, and provide relevant details of the HAWC observatory (our example experiment).

BHs can be completely characterised by their mass, charge and angular momentum. However, EBHs radiate charge and angular momentum faster than they radiate mass [68–73]. As such, we can assume that EBHs, at the end of their lives, are Schwarzschild black holes, which are uncharged and non-rotating. Schwarzschild BHs are then completely characterised by their mass, $M$.

Working in units where $\hbar = c = \kappa_{\mathrm{B}} = 1$, the temperature of a BH is given by [8,9]

$$T = \frac{1}{8\pi\,GM}\,, \tag{1}$$

where $G$ is the gravitational constant. BHs heavier than $\sim 10^{-8} M_\odot \sim 10^{26}\,$g are colder than the CMB and are absorbing CMB photons, so are gaining mass [74,75]. Lighter BHs, on the other hand, radiate particles of energy $E$ at the rate [8,9]

$$\frac{d^2 N_{\mathrm{p}}^i}{dt\,dE} = \frac{n_{\mathrm{dof}}^i \Gamma^i(M, E)}{2\pi(e^{E/T} \pm 1)}\,, \tag{2}$$

where $n_{\mathrm{dof}}^i$ is the number of degrees of freedom of particle $i$, $+$ $(-)$ corresponds to fermions (bosons) and $\Gamma^i(M, E)$ is a greybody factor that for a Schwarzschild black hole depends on the spin and energy of the radiated particle and on the mass of the black hole. The greybody factor can be calculated by solving a Schrödinger-like wave equation and finding the transmission coefficient of the solution from the BH horizon to infinity. We take the values made publicly available in the `BlackHawk` code [76], which we validated against the results in [67]. Intuitively, this greybody factor accounts for the fact that not all particles emitted by the black hole can escape its gravitational potential. Although it would be unphysical to do so, omitting the greybody factor would significantly increase the primary particle flux around $E \sim T$ for spin 1/2 and spin 1 particles, and so predict more observed photons for an EBH at a given distance.

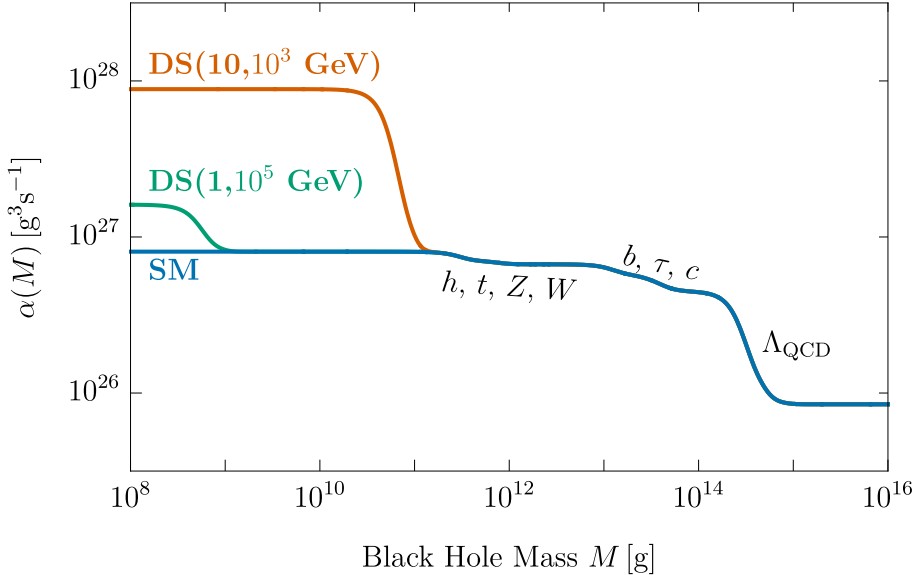

Figure 2: The function $\alpha(M)$, which accounts for all directly emitted particle species, for the SM and two dark sector models (see text for details). The SM particle labels show the particles responsible for the thresholds. Light quarks and gluons are only radiated above $\Lambda_{\rm QCD}$.

For $E \gg m^i$, where $m^i$ is the mass of the radiated particle, $\Gamma^i$ can be written as a function of the dimensionless quantity $x = 8\pi GME$. Although at $E \sim m^i$ there is a correction to this approximation [73], particles with $E \sim m^i$ only make up a small proportion of the radiated particles and we neglect this effect. At $E < m^i$, $\Gamma^i = 0$. The greybody factor then only depends on the particle spin and $x$. The primary photon spectra for a range of BH masses are shown in fig. 1.

Conservation of energy implies that as the BH radiates, it must lose mass. The BH mass evolves according to [72]

$$\frac{dM}{dt} = -\frac{\alpha(M)}{M^2},\tag{3}$$

where

$$\alpha(M) = M^2 \sum_i \int_0^\infty \frac{d^2N_{\rm p}}{dtdE}(M,E)E\,dE,\tag{4}$$

and the sum is over all particle species. All fundamental degrees of freedom present in nature with a de Broglie wavelength of the order of the black hole size are radiated [77], so contribute to $\alpha(M)$. Note in particular that $\alpha(M)$ is independent of the particle's non-gravitational interaction strengths. In fig. 2 we show $\alpha(M)$ for the SM in blue.

Although EBHs emit all particles, only stable particles can reach the earth to be observed, and only uncharged particles will be unaffected by the galaxy's magnetic field. Here we will focus on the photon spectrum of an EBH, which may be observed by a gamma ray observatory.

Primary photons are radiated directly from the EBH, according to eq. (2). Although these are emitted off-shell, the spectra of the final on-shell photons are very similar to the primary spectra. The other particles which are radiated may also produce secondary photons, as final state radiation or as the particles hadronise and decay. The secondary photon spectrum is given by the sum of the primary spectra integrated against the secondary spectrum of a primary

particle $i$ with energy $E_p$, $dN^{i \to \gamma}/dE$,

$$\frac{d^2 N_s^\gamma}{dt dE} = \sum_i \int_0^\infty \frac{d^2 N_p^i}{dt dE_p}(M, E_p) \frac{dN^{i \to \gamma}}{dE}(E_p, E) dE_p. \tag{5}$$

Computation of the secondary photon spectra is relatively complex, particularly in the case of coloured particles which hadronise. To calculate the secondary spectra we use the public code Pythia 8.3 [78], which we validated against BlackHawk [76]. The secondary photon spectra for several BH masses are shown in fig. 1.

Once produced, these photons then travel to the earth where they may be detected. The number of photons reaching the earth per $m^2$ will be reduced by the geometric factor $1/4\pi r^2$, where $r$ is the distance to the EBH. Although an EBH is yet to be observed, we investigate what information could be obtained if one were to be seen in a ground-based gamma ray observatory. As an illustrative example we take HAWC, the High Altitude Water Cherenkov Experiment located in Mexico at an altitude of 4100 meters, which started running in 2015. HAWC is ideally suited to this search as it has a high up-time fraction (95% [79]), large field of view (around 2 steradians [79]) and a large effective area ($\sim 10^5 \, m^2$ [80]). Due to these features, HAWC currently sets the strongest direct limits on the rate of EBHs at the parsec scale [81]. HAWC is sensitive to gamma rays from around 100 GeV to above $10^5$ GeV [80]. For gamma rays above $\sim 10^4$ GeV, HAWC has an effective area of $\sim 10^5 \, m^2$, but this falls off sharply at lower energies; at 100 GeV it is just $\sim 50 \, m^2$. The parameterisation of the effective area can be found in ref. [80]. Although we expect very few photons to be observed above $10^5$ GeV, we extrapolate the effective area from $10^5$ GeV to $10^7$ GeV, with a constant effective area. While photons with an energy above $\sim 10^5$ GeV may be converted into electron-positron pairs as they travel to earth, by interactions with cosmic microwave background photons and galactic interstellar light [82], this absorption is negligible for the parsec-scale distances considered in this work.

Unless the EBH occurs along the same line-of-sight as a powerful gamma ray source, the dominant background will come from cosmic rays which are misidentified as gamma rays. HAWCs cosmic ray rejection improves with energy and can reject more than 99.6% of cosmic ray showers with an energy greater than 10 TeV [83]. While this background could be taken into account with a detailed understanding of the detector, we opt for a conservative approach and restrict observation energies to be larger than $E_{min} = 10$ TeV and observation times to be less than $\tau_{max} = 1000$ s. With these cuts, less than one background event is expected within 1 degree of the EBH (a conservative estimate of the angular resolution of HAWC [80]). Other possible backgrounds are the extra-galactic [84] and galactic [85–90] gamma-ray backgrounds. However, we find these backgrounds to be smaller than the misidentified cosmic ray background, unless the EBH happens to line up with a gamma ray point source or the galactic centre. In making these comparisons we had to extrapolate some data to higher energies. When doing so, we assumed no drop in $E^2$ times the flux from the highest measured energy bin.

## 3 Probing the Dark Sector

To illustrate the sensitivity of an observation to BSM physics, we take the example of a dark sector (DS). As it is not known whether the DS communicates with the SM via interactions beyond the gravitational interaction, it is very difficult to conclusively probe these models in conventional dark matter experiments. However, since Hawking radiation is independent of these couplings, EBHs are uniquely placed to shine a light on the DS.

The DS could be simply a single dark matter particle, DS($\chi$) where we take $\chi$ to be a Dirac fermion, or could contain many more degrees of freedom, see e.g., refs. [91–98]. For illustrative purposes we consider models motivated by the Mirror Dark Matter [93] scenario, where the DS contains an exact copy of the SM degrees of freedom which communicate with the SM only via small portal couplings. Generalising [93], we will assume $N$ copies of the SM and take all particles in the dark sector to have a common mass, $\Lambda_{\mathrm{DS}}$. We will denote these models DS($N, \Lambda_{\mathrm{DS}}$). The function $\alpha$ for two benchmark models are shown in fig. 2. The increase in $\alpha$ at black hole masses $\sim 10^9 (10^{11})\,$g leads to an accelerated evaporation rate in the final $\sim 1 (10^6)\,$s of the BHs life. Since the DS particles will produce no (or very few) secondary photons, this acceleration will indicate the existence of the DS.

To distinguish SM evolution from BSM evolution at the HAWC observatory, we integrate the total photon spectra against the HAWC effective area over energies greater than $E_{\mathrm{min}} = 10^4\,$GeV and over intervals in the remaining lifetime of the EBH, $\tau$,

$$N_j = \frac{1}{4\pi r^2} \int_{E_{\mathrm{min}}}^{\infty} dE \int_{\tau_j}^{\tau_{j+1}} d\tau \frac{d^2 N_{\mathrm{p+s}}^{\gamma}}{d\tau dE} A(E, \theta, \tau), \tag{6}$$

where $A(E, \theta, \tau)$ is the effective area at zenith angle $\theta$ and time $\tau$, and $\tau_j \in \{10^{-4}, 10^{-3}, 10^{-2}, 10^{-1}, 10^0, 10^1, 10^2, 10^3\}$.[1] While this approach does not make use of the photon energy spectrum, we note that HAWC's energy resolution is relatively poor ($\sim 50\%$ for photons above $10^4\,$GeV). It does however make good use of the timing information, where HAWC has excellent resolution (order $100\,$ps). To approximate the motion of the EBH through the sky, we assume that the HAWC detector lies on the equator of the earth (it in fact lies at $19°$ N) and that the EBH occurs on the celestial equator. We also assume that the EBH spends its final 1000 seconds in the primary zenith angle band ($-26°$ to $26°$).

The integrated photon counts for the SM and DS models at different mass scales are shown in fig. 3, for an EBH seen at a distance of $0.01\,$pc. We see that more degrees of freedom lead to a lower photon count, due to the accelerated evaporation rate. We also see that a relatively light DS ($\Lambda_{\mathrm{DS}} \lesssim 10^3\,$GeV) leads to a reduction in the spectrum at all times, while a heavier DS ($\Lambda_{\mathrm{DS}} \sim 10^5\,$GeV) only alters the spectrum below $\tau \sim 1\,$s. This is because the EBH is only hot enough to emit such heavy particles in its last $1\,$s.

When an EBH is observed, however, its distance from earth will be unknown. If the SM is assumed, the total photon count can be used to determine the distance. Since here we are considering BSM models, we cannot make this assumption. Instead, we characterise the event by the total number of photons observed between $10^{-4}$ and $10^3\,$s. We then normalise the SM and BSM spectra, such as those presented in fig. 3, to yield this total photon count, as shown in fig. 4. We see that for $\Lambda_{\mathrm{DS}} \lesssim 10^3\,$GeV, there are relatively fewer photons at $\tau > 10^2\,$s. This is because the EBH is cooler than in the SM scenario at that time, so fewer photons above $E_{\mathrm{min}}$ are emitted. For heavier mass scales, such as $\Lambda_{\mathrm{DS}} = 10^5\,$GeV, there are relatively more photons at $\tau > 10^2\,$s. For $\Lambda_{\mathrm{DS}} \sim 10^4\,$GeV the normalised spectrum is very similar to the SM spectrum, so we expect a loss of sensitivity in this region.

After normalising the spectra, we perform a chi-squared test between the expected observed spectrum (given by the SM) and the BSM spectra. We add the statistical and systematic errors in each bin in quadrature. Figure 5 shows the expected $2\sigma$ limits that could be placed on various DS models for different DS mass scales and different systematic errors. The left axis gives the total number of photons observed between $10^{-4}$ and $10^3\,$s with $E > E_{\mathrm{min}} = 10^4\,$GeV, while the right axis gives the inferred distance to the EBH assuming only the SM.[2] If the local

---

[1]Only a small percentage of the total photons received will be received after $\tau \sim 10^{-4}\,$s, while a further bin up to $10^4\,$s would contain significant background from misidentified cosmic rays.

[2]If a non-SM signal is observed, the distance to the evaporating black hole cannot be inferred by a simple

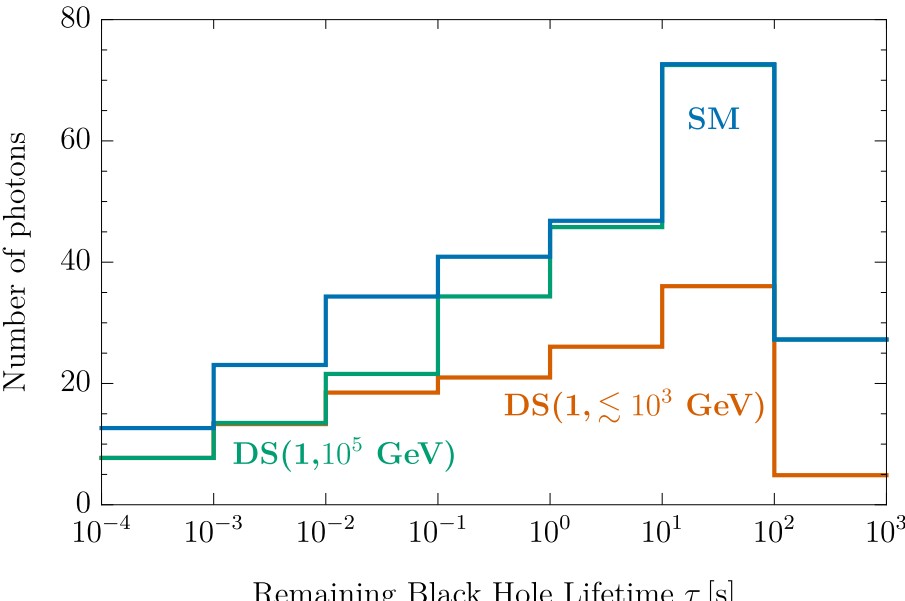

Figure 3: The number of photons observed in each time window for an EBH observed at 0.01 pc for the SM and dark sector models at different mass scales.

EBH density is near the current upper limit ($3400 \, \mathrm{pc}^{-3} \, \mathrm{yr}^{-1}$ [81]), the probability of HAWC observing at least one event in the next five years at a distance less than 0.05 (0.01) pc is $\sim 83\%$ (1.4%), which corresponds to observation of around 10 (200) photons.

We see that when there are more degrees of freedom in the DS, fewer photons are required to exclude the model. DS(100) can be essentially excluded up to $10^5$ GeV with $\mathcal{O}(10)$ photons, while DS(1) requires $\sim 200$ photons.

For a dark sector mass scale $\lesssim 2$ TeV, the new radiation processes have fully opened by $\tau \sim 10^3$ s. Since this is the total length of assumed observation time, the search becomes independent of the mass scale below $\sim 2$ TeV. At mass scales $\gtrsim 10^6$ GeV, the search loses sensitivity since the EBH emits very few particles at such high energies. As mentioned above, there is a loss of sensitivity around $\Lambda_{\mathrm{DS}} = 10^4$ GeV, due to an interplay between $E_{\min}$ and the maximum time window. This region could be better probed by relaxing these cuts and accurately incorporating the cosmic ray background into the analysis.

In the lower half of the plot, the exclusion limit is dominated by statistical errors and the limit does not significantly change for $\sigma_{\mathrm{syst.}} \lesssim 5\%$. In the top half of the plot, so many photons are received that the systematic error has a significant impact on the limit. We see that $\sigma_{\mathrm{syst.}} \lesssim 1\%$ is required to place good limits on the DS($\chi$) model.

## 4 Discussion

In this work we have considered the limits that could be placed on DS models by the observation of an EBH by the HAWC observatory. HAWC has a large field of view (around 2 steradians) and up-time fraction (95%) [79], as well as good timing resolution (order 100 ps) and reasonable angular resolution ($0.1 - 1$ degree). These attributes mean that it is well suited to

measurement of the photon count with time, as this measurement contains a degeneracy between the distance and the underlying particle physics model. Analysis of the photon energy spectrum and/or multi-messenger searches could potentially be leveraged to break this degeneracy. Also, if one assumes that new particles are only present above a certain scale, the spectrum below that scale could be used to calibrate the distance.

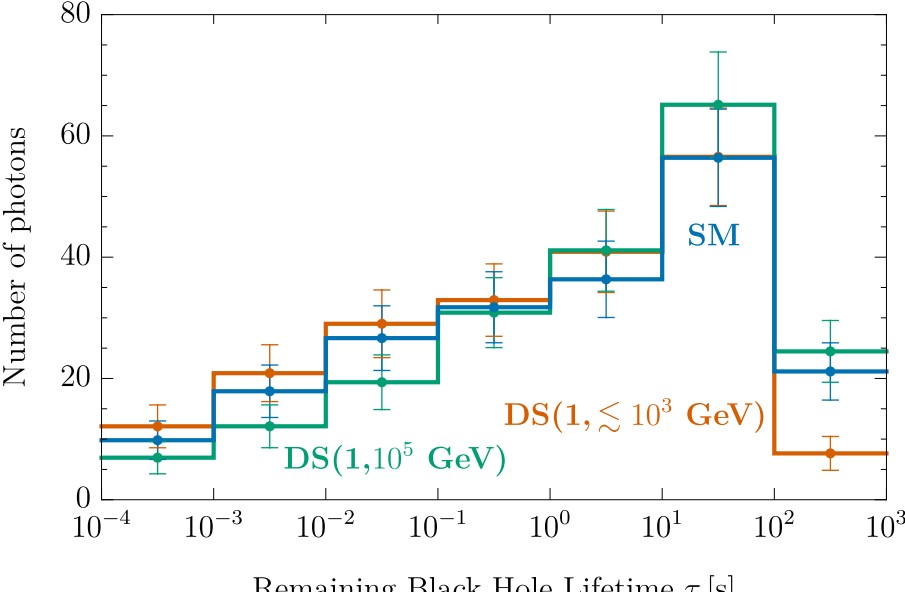

Figure 4: The number of photons observed in each time window normalised to 200 total photons, for the SM and dark sector models at different mass scales. The error bars include statistical and 5% systematic errors.

observing an EBH by chance. However, it has relatively poor energy resolution $(30 - 100\%)$, so the present analysis does not make use of this information. Other experiments, such as HESS, have better energy resolution so could make use of the gamma-ray energies, but they would be very unlikely to be pointing at an EBH by chance. As such, an early warning would need to be provided from another experiment. It would be interesting to investigate how this could best be done for a variety of BSM scenarios, given that it takes around 100 s to re-position a telescope like HESS. LHAASO has also recently begun taking data and while in many ways it performs similarly to HAWC, its muon detection capabilities mean that its cosmic ray rejection rate is around two orders of magnitude better than HAWC's at energies above 10 TeV [99–101]. However, it is not clear that this gives LHAASO a clear advantage in this case since the EBH only emits photons above 10 TeV in its final $10^3$ s, and HAWC has zero background over this time period. Fermi-LAT has also performed a search for EBHs within $\sim 0.03$ pc [102], obtaining an upper limit of 7200 pc$^{-3}$ yr$^{-1}$, which is comparable to the upper limit from HAWC. However, in the event of an observation, Fermi-LAT would not be able to probe models of new physics above $\sim 100$ GeV since its small area ($\sim 1$ m$^2$) means it requires a long integration time ($\sim 10^8$ s) to obtain enough photons. As such, it would not detect a significant number of photons from the final burst, where high scale new physics models are probed. In the future it would be very interesting to investigate whether high-energy neutrinos emitted from the EBH could be observed at, e.g., Ice-Cube and could be correlated with the gamma-ray signal. Similarly, the event could potentially also be correlated with an anti-proton or positron burst.

While we have demonstrated sensitivity to models with mass scales below $\sim 10^6$ GeV, one could imagine that this limit could be raised in the future. EBHs continue to radiate particles at least up to the Planck scale, and the experimental timing resolution of HAWC allows for measurement down to $\sim 100$ ps (in principle probing masses up to $\sim 10^8$ GeV). However, we see from fig. 3 that the number of photons received at HAWC significantly reduces in the bins as $\tau \to 0$. Even though the flux is increasing as the EBH shrinks (see fig. 1), the shorter time window results in a lower photon count. Extending the present analysis to $\tau < 10^{-4}$ s

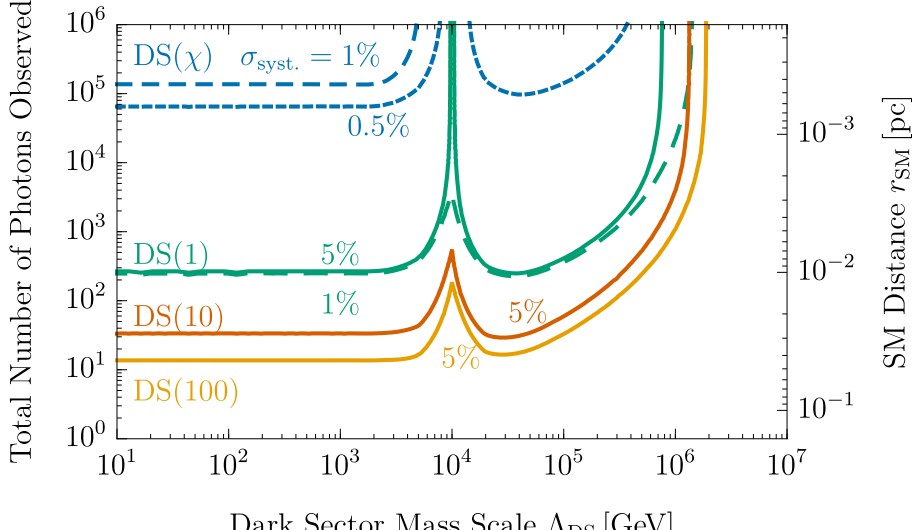

Figure 5: Projected $2\sigma$ exclusion limits for a range of dark sector models, for different systematic errors. The search assumes that a given total number of photons is observed between $10^{-4}$ and $10^3$ s, with a SM-like spectrum. The distance to the EBH, assuming only the SM, is given by the right axis.

would only provide a few percent more photons, which would not significantly improve the sensitivity of this analysis (which does not make use of the energy of each photon detected). However, future experiments with better energy resolution could make use of this information and look in detail at the few highest energy photons seen at the very end of the explosion. This may then provide a probe of new physics scales above $\sim 10^6$ GeV. While the use of Pythia 8.3 is sufficient for the present analysis (and useful since we cross-check our results against BlackHawk), it does not accurately model the secondary spectra above $10^7$ GeV. Software such as [103], which is designed to be applicable to Planck scale energies, could be used in these future investigations.

As a note of caution, the expected chance of observing an EBH in the near future remains uncertain. While the probabilities given above assume the upper limit of the local BH burst rate (3400 pc$^{-3}$ yr$^{-1}$ [81]), other constraints may be significantly stronger. However, the applicability of these constraints is debated. Although the limits from galactic and extra-galactic $\sim 100$ MeV gamma-ray signals indeed seem strong, 0.06 pc$^{-3}$yr$^{-1}$ and $10^{-6}$ pc$^{-3}$yr$^{-1}$ respectively [38, 104–106], the extrapolation from these indirect constraints to the probability of observing an EBH relies on several assumptions. For the extra-galactic limits, clustering of BHs in galactic halos, which is expected, reduces the limit from $10^{-6}$ pc$^{-3}$yr$^{-1}$ to 10 pc$^{-3}$yr$^{-1}$ [38, 104, 105]. For both the galactic and extra-galactic limits, there has been a long debate in the literature about the BH emission around 100 MeV, for instance due to details of the QCD phase transition or the formation of an optically thick photosphere. A recent review concludes that perhaps the best strategy is to accept that our understanding of such effects is incomplete and to focus on the empirical aspects of the $\gamma$-ray burst observations [38]. The other main indirect constraints come from cosmic rays (primarily anti-protons), which set a limit around $10^{-3}$ pc$^{-3}$yr$^{-1}$ [107]. However, these constraints depend sensitively on models of both the production of secondary anti-protons via interactions with interstellar gas and the propagation of primary and secondary anti-protons through the galaxy [80]. In summary, while there are a range of possible constraints, none of them are currently secure enough to conclude that an EBH will not be observed in the near future and the most robust limit on our scenario is ref. [81].

## 5  Conclusions

The observation of an EBH can place significant constraints on the number of elementary degrees of freedom present in nature. We have exemplified this with a variety of dark sector models, and found that the number of new degrees of freedom below $\sim 10^5$ GeV could conceivably be limited to less than one copy of the SM degrees of freedom in the near future.

The approach outlined here could readily be extended to further BSM models, in particular those with large numbers of new degrees of freedom. Given that such an observation can probe mass scales up to $\gtrsim 10^6$ GeV, models which address the hierarchy problem, such as SUSY, composite Higgs models and NNaturalness [108], would be of particular interest. Other interesting scenarios would be light new physics sectors, where the non-gravitational interaction strengths are typically very weak, or further models with large numbers of new particles such as extra dimensional models with towers of KK resonances or string theory (which often leads to an abundance of light scalar particles). In contrast to the dark sector models considered here, some of these new particles will produce additional secondary photons, which may improve the sensitivity of both the initial EBH search and the information that can be extracted from the signal.

However, as we have demonstrated, the information obtained from an observation would be unique and of fundamental importance. While we have considered five years of observation by the HAWC observatory, improved experiments such as CTA [109] and SGSO [110] are in development. The larger effective area of these experiments significantly increases the potential observation rate, and improved energy resolution could help determine the distance to an EBH even in BSM scenarios. Furthermore, multiple experiments could potentially observe the same event (at similar or lower photon energies), and multi-messenger approaches could possibly see the event in other particles, such as neutrinos.

## Acknowledgements

The authors would like to thank Peter Skands for advice on using `Pythia 8.3`, and would like to acknowledge support from the Australian Government through the Australian Research Council.

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
