# Peer review of "Probing the Particle Spectrum of Nature with Evaporating Black Holes"

_SciPost Physics, doi:SciPost Phys. 12, 150 (2022)_

## Round 2 · Referee Report · Anonymous (Referee 2) · 2022-2-17

Report

I appreciate the thorough revisions of the authors and recommend the manuscript for publication. Below are my comments on the main points.

  1. I very much appreciated the extended discussions regarding the probability for HAWC to observe an EBH. While the likelihood remains low, I certainly think it is a possibility to be prepared for, and feel the manuscript is significantly strengthened by a clear discussion of what other constraints might be at present and where loopholes to those could exist.

  2. I found the new justification for using Pythia clear.

  3. & 4. I’m glad the authors looked into this carefully, and I am much more confident in the updated analysis given there is a more careful discussion of the possible backgrounds.

  4. & 6. I found the discussion of alternative experiments helpful and also convincing as to why they focussed on HAWC. One point I would note, however, is that in addition to its improved cosmic ray rejection rate, LHAASO also has a much larger effective area than HAWC, which the authors don’t mention, and I would imagine this is strictly beneficial for such searches.

I also appreciated the changes made in response to my minor comments. To briefly respond to point 3, however, I’m not convinced by what the authors wrote. If there is a process where a photon could be emitted, I imagine there can also be a higher order process where that same photon split, for instance. Further, if it was the case that the emitted states should be treated as on-shell, then why would the authors include, for example, electrons in the sum, as on-shell electrons do not emit photons through final state radiation. (Indeed, the hard objects emerging from a collision in Pythia are not on-shell, and yet this is what the authors use to determine other aspects of the spectrum.) Regardless, I do not think this is a major point as the inclusion of photons in the sum will not make an appreciable difference to the results, but if the authors could clarify this point I suspect it would avoid any confusion around this propagating forward.

  • validity: high
  • significance: good
  • originality: good
  • clarity: good
  • formatting: perfect
  • grammar: perfect

Author:  Michael J. Baker  on 2022-05-03  [id 2429]

(in reply to Report 1 on 2022-02-17)

We thank the referee for their comments.

Regarding the photon splitting, we now agree with the referee and accept that we were incorrect in our previous response. The photons are indeed emitted off-shell and we model them as such. While we used the primary spectra in our analysis, we have verified that these secondary photons are subdominant to the main secondary spectra from the other primary particles and that the primary photon spectra reduce by at most 6% due to the splitting. We have updated the manuscript to reflect this. We thank the referee for asking us to look at this again.

---

## Round 2 · Referee Report · Anonymous (Referee 1) · 2022-3-22

Report

I appreciate the authors adding further clarifying discussions related to detection prospects as well as cross-checking with BlackHawk code.

Regarding expected event rate and probability of such BHs, the authors assumed HAWC upper limit based on local emission. As the authors show on their Fig. 2, the effects they are interested in appear to become relevant for BHs with masses below 10^(11)g. At these masses, the BH lifetime is less than 1 year. Such BHs would have evaporated by now if they were to form in the early Universe (i.e. primordial). What kind of viable mechanisms do the authors consider for producing or motivating such extremely late evaporating non-primordial BHs and what is their relation with dark matter abundance?

Related to this, the authors comment on existing constraints for evaporating BHs. However, some of these constraints have been calculated under assumption of DM abundance and primordial BHs. Hence, it is difficult for me to see how the authors can discuss this without clarifying relation of their scenario to dark matter abundance and formation mechanisms.

I would like to ask the authors to further comment on these issues in the paper.
  • validity: -
  • significance: -
  • originality: -
  • clarity: -
  • formatting: -
  • grammar: -

Author:  Michael J. Baker  on 2022-04-27  [id 2420]

(in reply to Report 2 on 2022-03-22)

We thank the referee for their comments, which we address here as requested by the Editor-in-charge.

While we are agnostic as to the origin of the evaporating black holes, we have in mind approximately 10^15 g black holes which originated in the early Universe and slowly lost mass due to radiation until their final explosion today. The main observational constraint on this population of black holes would be 100 MeV photons, and we address this issue in the discussion.

In this work we make no assumptions about dark matter. While PBHs could possibly constitute dark matter, we do not make this assumption in this work. While we are studying a dark sector, we also do not require that dark matter is the lightest dark sector particle. While this could provide some motivation for the models we consider, it is not a requirement for the analysis we present.

As far as we are aware, none of the constraints we discuss require dark matter to be constituted of PBHs. They instead place limits on the density of small or exploding black holes (which is often expressed as a fraction of the dark matter abundance).

---

## Round 2 · List of Changes

Updated to correctly account for dominant background, conclusions unchanged.

Expanded discussion of other current and future experiments, and of limits on the chance of observation.

---

## Editorial Decision

published